# $Z_2$ order fractionalization, topological phase transition, and odd frequency pairing in an exactly solvable spin-charge ladder

Jian-Jian Miao[1,2] and Wei-Qiang Chen[2]

[1]*Quantum Science Center of Guangdong-Hong Kong-Macao Greater Bay Area (Guangdong), Shenzhen 508045, China*
[2]*Department of Physics and Shenzhen Key Laboratory of Advanced Quantum Functional Materials and Devices,*
*Southern University of Science and Technology, Shenzhen 518055, China*
(Dated: May 22, 2024)

Motivated by the order fractionalization in Kitaev-Kondo model, we propose an exactly solvable spin-charge ladder model to study the order fractionalization with discrete symmetry. The spin-charge ladder is composed of a spin chain and a superconducting wire coupled via an Ising-type interaction, and we obtain the exact solution in the flat band limit. The exact solution reveals the $Z_2$ order fractionalization with dual symmetry breaking and intertwined order parameters. We investigate the topological phase transition of the spin-charge ladder via the spectral chiral index, and identify the correlated topological superconductor (TSC*) phase with gapped $Z_2$ Kondo flux excitations. We demonstrate Majorana spinons generated odd frequency pairing in the superconducting wire. We also discuss the order fractionalization in the perspective of $Z_2$ lattice gauge theory.

## I. INTRODUCTION

Fractionalization is an intriguing concept in theoretical physics and plays a vital role in understanding the exotic phenomenon in strongly correlated systems, such as the quasiparticles with fractional charge and fractional statistics in fractional quantum Hall effect[1,2]. In the development of the theory of high-temperature superconductivity, the perspective of fractionalization provides profound insights. In the slave-particle theory, the electron is fractionalized into spinon and holon carrying the charge and spin degrees of freedom respectively[3]. However, the slave-particle theory inevitably introduces gauge field and the corresponding lattice gauge theory with continuous gauge symmetry suffers from the confinement problem[4] that can invalid the spin-charge separation scenario. A different $Z_2$ gauge theory of electron fractionalization with discrete $Z_2$ gauge symmetry is proposed to implement the deconfined insulating phase[5]. The gapped vortex excitations, i.e. visons, of the $Z_2$ gauge field ensure the insulating phase is fractionalized. The milestone is the celebrated Kitaev's honeycomb model[6], an exactly solvable model manifesting Majorana fractionalization of spins. Such an exact solution provides a solid foundation for the theoretical framework of fractionalization.

Along the lines of Majorana fractionalization of spins, Tsvelik and Coleman propose a novel concept termed order fractionalization in the Kitaev-Kondo model[7]. Started with the generalized Kitaev's honeycomb model with $SU(2)$ spin symmetry, i.e. the Yao-Lee model[8], the low-energy excitations of the quantum spin liquid are gapless Majorana spinons and gapped visons. By coupling the Yao-Lee spin liquid to the conduction electron via the Kondo interaction, they explore the possibility that a fractionalized Majorana spinon and an electron form a bound state. The composite boson inherits fractional quantum numbers from the Majorana spinon and electron. The condensation of the composite boson gives rise to the order fractionalization with fractionalized order parameters. The order fractionalization neither like conventional order as its fractional quantum numbers, nor like topological order as its symmetry breaking. The mean-field theory of Kitaev-Kondo model in two dimensions and random phase approximation theory of Coleman-Panigrahi-Tsvelik model[9] in three dimensions demonstrate the rich phenonmenon in order fractionalization, such as pair density wave and odd-frequency pairing.

In this paper, we propose a spin-charge ladder model composed of a spin chain and a superconducting wire. We obtain the exact solution of the spin-charge ladder in the flat band limit. With the help of the exact solution, we study the $Z_2$ order fractionalization therein and derive the exact results about dual symmetry breaking and intertwined order parameters. In the perspective of Majorana-SSH model, we explore the topological phase transition of the spin-charge ladder. In the low-energy, we clearly demonstrate that Majorana spinons effectively generate odd frequency pairing in the superconducting wire. The current study offers valuable exact results about order fractionalization for future study.

The paper is organized as follows. First, in Sec. II, we introduce the model Hamiltonian of the spin-charge ladder. In Sec. III, we study the spin chain in combination of Majorana fractionalization and Jordan-Wigner transformation. We propose the Majorana-SSH model to understand the topological phase transition of the spin chain. In Sec. IV, we employ the perspective of Majorana-SSH model to study the superconducting wire. In Sec. V, we obtain the exact solution of the spin-charge ladder in the flat band limit, and explore the $Z_2$ order fractionalization, topological phase transition, and odd frequency pairing. Finally, we summarize the results and discuss the case away from the flat band limit in Sec. VI.

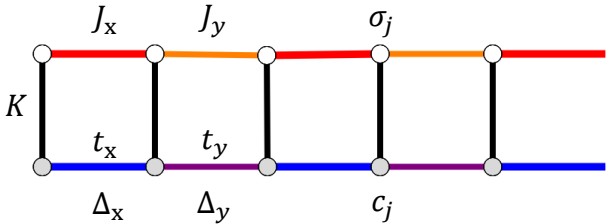

FIG. 1: Schematic of the spin-charge ladder. The white and grey points denote the spin and charge degrees of freedom. $J_x$ and $J_y$ are the coupling constants of alternating Ising interactions of the spin chain. $t_x$, $t_y$ and $\Delta_x$, $\Delta_y$ are the alternating hopping matrix elements and pairing potentials respectively of the superconducting wire. $K$ is the coupling constant of the Ising-type interaction between the spin chain and superconducting wire.

## II.  MODEL HAMILTONIAN

Consider a lattice model describing a spin chain coupled to a spinless superconducting wire via an Ising-type interaction, the full Hamiltonian consists of three parts

$$H = H_S + H_C + H_I, \tag{1}$$

$$H_S = \sum_j \left( J_x \sigma_{2j-1}^x \sigma_{2j}^x + J_y \sigma_{2j}^y \sigma_{2j+1}^y \right), \tag{2}$$

$$H_C = \sum_j \left( t_x c_{2j-1}^\dagger c_{2j} + t_y c_{2j}^\dagger c_{2j+1} + h.c. \right)$$
$$+ \sum_j \left( \Delta_x c_{2j-1}^\dagger c_{2j}^\dagger + \Delta_y c_{2j}^\dagger c_{2j+1}^\dagger + h.c. \right), \tag{3}$$

$$H_I = \sum_j K \sigma_j^z \left( c_j^\dagger c_j - c_j c_j^\dagger \right), \tag{4}$$

where $j$ denotes the lattice site. $\sigma_j^\alpha$ ($\alpha = x, y, z$) are Pauli matrices denoting the spin degrees of freedom. $H_S$ describes a spin chain with alternating Ising interactions in the $x$ and $y$-directions, and the coupling constants are $J_x$ and $J_y$ respectively. The spin chain can be viewed as the one-dimensional limit of the Kitaev's honeycomb model[6]. $c_j^\dagger$ and $c_j$ are the creation and annihilation operators of spinless fermions or spin polarized electrons denoting the charge degrees of freedom. $H_C$ describes a spinless superconductor wire with dimerized hopping and pairing, the hopping matrix elements are $t_x$ and $t_y$, and the pairing potentials are $\Delta_x$ and $\Delta_y$. The hopping part of $H_C$ is the spinless Su-Schrieffer-Heeger (SSH) model[10], and $H_C$ is the dimerized Kitaev chain[11]. $H_I$ is an Ising-type interaction that couples the spin and charge degrees of freedom with the coupling constant $K$. $c_j^\dagger c_j - c_j c_j^\dagger = 2(c_j^\dagger c_j - 1/2) = \pm 1$ is the charge density measured with respect to half-filling that features the Ising characteristic. $H_I$ resembles the $z$-component of the Kondo coupling. The above toy model extracts the three essential elements of the Kitaev-Kondo model,

i.e. spin, fermion and coupling, to realize the order fractionalization, and is called spin-charge ladder as shown in FIG. 1.

## III.  SPIN CHAIN: MAJORANA-SSH MODEL

We first consider the decoupled limit and study the spin chain and superconducting wire separately, then adopt the consistent perspective to investigate the spin-charge ladder. The spin operators can be represented by Majorana fermion operators and we shall employ two Majorana fermion representations. The first *local* Majorana fermion representation represents the spin operators in terms of four Majorana fermions locally through the Majorana fractionalization

$$\sigma_j^x = i\gamma_j^x \gamma_j^z, \tag{5}$$
$$\sigma_j^y = i\gamma_j^x \gamma_j^t, \tag{6}$$
$$\sigma_j^z = i\gamma_j^x \gamma_j^y, \tag{7}$$

where $\gamma_j^\alpha (\alpha = x, y, z, t)$ are Majorana fermion operators satisfying the Clifford algebra

$$\left( \gamma_j^\alpha \right)^\dagger = \gamma_j^\alpha, \tag{8}$$
$$\{ \gamma_j^\alpha, \gamma_l^\beta \} = 2\delta^{\alpha\beta} \delta_{jl}. \tag{9}$$

As the quantum dimension of single Majorana fermion operator is $\sqrt{2}$, four Majorana fermions enlarge the local two-dimensional spin Hilbert space to four-dimensional enlarged Hilbert space. To faithfully represent the spin degrees of freedom, one method is using the projection operator to eliminate the redundant degrees of freedom. In consideration of the local identities in the spin Hilbert space

$$\sigma_j^x \sigma_j^y \sigma_j^z = i, \tag{10}$$

the local constraints are imposed

$$\hat{D}_j \equiv \gamma_j^x \gamma_j^y \gamma_j^z \gamma_j^t = 1, \tag{11}$$

which halves the dimensions of the enlarged Hilbert space. The spin wavefunction can be obtained from the Majorana fermion wavefunction through the Gutzwiller projection[12]

$$|\psi_{\text{spin}}\rangle = \hat{P}_G |\psi_{\text{Majorana}}\rangle, \tag{12}$$

where $\hat{P}_G = \prod_j (\hat{D}_j + 1)/2$ is the projection operator. Another method is treating the redundant degrees of freedom as gauge degrees of freedom. The $Z_2$ gauge operator $\hat{D}_j$ implements the local $Z_2$ gauge transformation

$$\hat{D}_j \gamma_j^\alpha \hat{D}_j = -\gamma_j^\alpha. \tag{13}$$

The spin wavefunction is the equal weight linear superposition of gauge equivalent Majorana fermion wavefunctions. Therefore the Majorana fractionalization of spins

renders the spin model into $Z_2$ lattice gauge theory. The Hamiltonian of the spin chain in the local Majorana fermion representation is given by

$$H_S = \sum_j \left( J_x \hat{u}_{2j-1} i\gamma^x_{2j-1}\gamma^x_{2j} - J_y \hat{u}_{2j} i\gamma^x_{2j}\gamma^x_{2j+1} \right), \quad (14)$$

where

$$\hat{u}_{2j-1} = -i\gamma^z_{2j-1}\gamma^z_{2j}, \quad (15)$$

$$\hat{u}_{2j} = i\gamma^t_{2j}\gamma^t_{2j+1}. \quad (16)$$

The operators $\hat{u}_j$ satisfy $\hat{u}_j^2 = 1$, and have $Z_2$ eigenvalues $u_j = \pm 1$. Under local $Z_2$ gauge transformation, the operators $\hat{u}_j$ transform as

$$\hat{u}_j \rightarrow \Lambda_j \hat{u}_j \Lambda_{j+1}, \quad (17)$$

where $\Lambda_j = \pm 1$. Hence the operators $\hat{u}_j$ are called $Z_2$ gauge field. We can choose the axial gauge

$$\hat{u}_j = 1, \quad (18)$$

to simplify the following calculations. In one-dimensional chain, the axial gauge fixes all the redundant gauge degrees of freedom. To justify the axial gauge, we introduce the second *nonlocal* Majorana fermion representation that represents the spin operators in terms of Majorana fermions nonlocally through the Jordan-Wigner transformation

$$\sigma^x_{2j-1} = \gamma^y_{2j-1} \prod_{l<2j-1} i\gamma^x_l\gamma^y_l, \quad (19)$$

$$\sigma^y_{2j-1} = \gamma^x_{2j-1} \prod_{l<2j-1} i\gamma^x_l\gamma^y_l, \quad (20)$$

$$\sigma^x_{2j} = -\gamma^x_{2j} \prod_{l<2j} i\gamma^x_l\gamma^y_l, \quad (21)$$

$$\sigma^y_{2j} = \gamma^y_{2j} \prod_{l<2j} i\gamma^x_l\gamma^y_l, \quad (22)$$

$$\sigma^z_j = i\gamma^x_j\gamma^y_j, \quad (23)$$

where different conventions on two sublattices are chosen to obtain the concise form of the Hamiltonian $H_S$. The Hamiltonian of the spin chain in the nonlocal Majorana fermion representation is given by

$$H_S = \sum_j \left( J_x i\gamma^x_{2j-1}\gamma^x_{2j} - J_y i\gamma^x_{2j}\gamma^x_{2j+1} \right), \quad (24)$$

which is the same as local Majorana fermion representation in the axial gauge. Thus the covention of nonlocal Jordan-Wigner string is equivalent to the gauge fixing. On average, one spin is represented by two Majorana fermions in the nonlocal Majorana fermion representation. Therefore the dimension of the spin Hilbert space is the same as the Majorana fermion Hilbert space. We can directly retrospect the spin degrees of freedom from the inverse Jordan-Wigner transformation. Meanwhile the

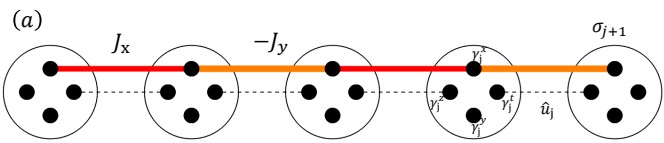

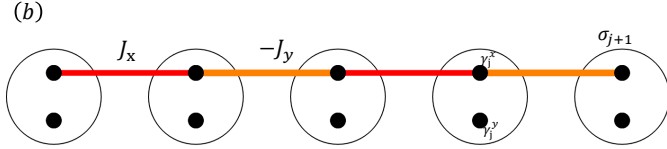

FIG. 2: The local (a) and nonlocal (b) Majorana representations of the spin chain. Each white circle denotes one spin, and each black point denotes one Majorana fermion. The dashed lines denote the $Z_2$ gauge field $\hat{u}_j$. The alternating solid lines connecting the $\gamma^x_j$'s constitute the Majorana-SSH model.

local Majorana fermion representation can be straightforwardly applied to higher dimensional systems for later study. We shall combine the local and nonlocal Majorana fermion representations to complement each other's advantages throughout the paper.

We shall investigate the spin chain in the perspective of *Majorana-SSH model*. Note the Hamiltonian $H_S$ in Eq. (24) has the dimerized form of Majorana fermions, similar to the dimerized form of complex fermions in the SSH model, therefore is named Majorana-SSH model. We perform the Fourier transformation of Majorana fermion operators

$$\gamma^x_{r\mu} = \sqrt{\frac{2}{N_c}} \sum_k e^{ikr}\gamma^x_{k\mu}, \quad (25)$$

where $r$ denotes the unit cell containing two sites from $A$ and $B$ sublattices, $N_c$ is the number of unit cells, and $\mu = A, B$. The operators $\gamma^x_{k\mu}$ in momentum space satisfy

$$\left(\gamma^x_{k\mu}\right)^\dagger = \gamma^x_{-k\mu}, \quad (26)$$

$$\left\{\gamma^x_{k\mu}, \left(\gamma^x_{p\nu}\right)^\dagger\right\} = \delta_{kp}\delta_{\mu\nu}. \quad (27)$$

Therefore $\gamma^x_{k\mu}$ are complex fermion operators defined in the half of the Brillouin zone. Nonetheless, it is more convenient to duplicate the operators $\gamma^x_{k\mu}$ in the whole Brillouin zone just like the redundant Bogoliubov-de-Gennes formalism of superconductors. In the momentum space, the Hamiltonian of the spin chain is

$$H_S = \sum_k \Gamma^\dagger_k \hat{h}_S(k) \Gamma_k, \quad (28)$$

$$\hat{h}_S(k) = \vec{d}_k \cdot \vec{\tau}, \quad (29)$$

where the imaginary unit $i$ is absorbed into the definition of the two-component vector $\Gamma_k = \begin{pmatrix} \gamma^x_{kA} & i\gamma^x_{kB} \end{pmatrix}^T$,

$\tau^{\alpha}$ $(\alpha = x, y, z)$ are Pauli matrices acting on the sublattice degrees of freedom, and

$$d_k^x = J_x + J_y \cos k, \tag{30}$$
$$d_k^y = J_y \sin k, \tag{31}$$
$$d_k^z = 0. \tag{32}$$

Note the momentum space Hamiltonian $\hat{h}_S(k)$ has the same form of the SSH model. With the sublattice chiral symmetry

$$\tau^z \hat{h}_S(k) \tau^z = -\hat{h}_S(k), \tag{33}$$

the topological invariant characterizing the bulk topology is the winding number

$$\begin{aligned}
\nu_{\gamma^x} &= \frac{1}{2\pi} \int_{-\pi}^{\pi} dk \big( \hat{d}_k \times \frac{d}{dk} \hat{d}_k \cdot \hat{z} \big) \\
&= -\frac{1}{2\pi i} \int_{-\pi}^{\pi} dk \frac{d}{dk} \ln\big( d_k^x - i d_k^y \big) \\
&= \Theta\big( J_y^2 - J_x^2 \big),
\end{aligned} \tag{34}$$

where $\hat{d}_k = \vec{d}_k / |\vec{d}_k|$ is a unit vector, and

$$\Theta(x) = \begin{cases} 1, & x > 0 \\ 0, & x < 0 \end{cases} \tag{35}$$

is the step function. Based on the bulk-edge correspondence, the winding number counts the number of zero modes localized on the edge. For the Majorana-SSH model, the Majorana fermions tend to pair on the strong bonds. For $|J_y| > |J_x|$, one $\gamma^x$ Majorana fermion is left unpaired on the edge. Therefore $\nu_{\gamma^x}$ counts the number of Majorana zero modes. The winding number indicates topological phase transitions at $J_x = \pm J_y$. The energy spectrum of the spin chain is

$$\epsilon_k = \pm|\vec{d}_k| = \pm\sqrt{\big(J_x + J_y \cos k\big)^2 + \big(J_y \sin k\big)^2}. \tag{36}$$

Note the gap also closes at quantum critical points $J_x = \pm J_y$, and is consistent with the results from winding number. All in all, the Majorana-SSH model has the same properties as the SSH model except the degrees of freedom halve.

The spin chain is studied previously by the Jordan-Wigner transformation and spin dual transformation[13]. The topological phase transition is characterized by the string order parameters. The Majorana-SSH model not only provides another perspective of the spin chain, but also paves the way for the study of the spin-charge ladder.

## IV. SUPERCONDUCTING WIRE: DECOUPLED MAJORANA-SSH MODELS

We shall also investigate the superconducting wire in the Majorana fermion representation. We decompose one

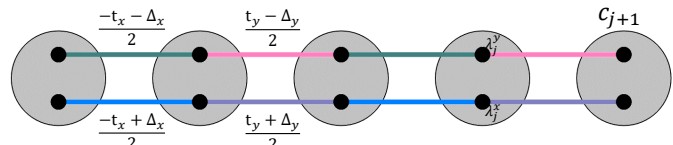

FIG. 3: The Majorana representation of the superconducting wire. Each grey circle denotes one charge, and each black point denotes one Majorana fermion. The superconducting wire is composed of two decoupled Majorana-SSH models.

complex fermion $c_j$ into two Majorana fermions $\lambda_j^x$ and $\lambda_j^y$ as follows

$$c_{2j-1} = \frac{1}{2}\big(\lambda_{2j-1}^x + i\lambda_{2j-1}^y\big), \tag{37}$$
$$c_{2j} = \frac{1}{2}\big(\lambda_{2j}^y - i\lambda_{2j}^x\big), \tag{38}$$

where different conventions on two sublattices are chosen to obtain the concise form of the Hamiltonian $H_C$. In the Majorana fermion representation, the Hamiltonian of the superconducting wire is given by

$$\begin{aligned}
H_C = \sum_j \big( \frac{-t_x + \Delta_x}{2} i\lambda_{2j-1}^x \lambda_{2j}^x + \frac{t_y + \Delta_y}{2} i\lambda_{2j}^x \lambda_{2j+1}^x \big) \\
+ \sum_j \big( \frac{-t_x - \Delta_x}{2} i\lambda_{2j-1}^y \lambda_{2j}^y + \frac{t_y - \Delta_y}{2} i\lambda_{2j}^y \lambda_{2j+1}^y \big),
\end{aligned} \tag{39}$$

which can be viewed as two decoupled Majorana-SSH models as shown in FIG. 3.

The properties of the superconducting wire can be obtained straightforwardly from the Majorana-SSH model. We introduce the total winding number $\nu_\lambda$ to characterize the bulk topology of the superconducting wire, which is defined as the sum of winding numbers of $\lambda^x$ and $\lambda^y$ Majorana fermions

$$\nu_\lambda = \nu_{\lambda^x} + \nu_{\lambda^y}, \tag{40}$$
$$\nu_{\lambda^x} = \Theta\big[(t_y + \Delta_y)^2 - (t_x - \Delta_x)^2\big], \tag{41}$$
$$\nu_{\lambda^y} = \Theta\big[(t_y - \Delta_y)^2 - (t_x + \Delta_x)^2\big]. \tag{42}$$

The total winding number counts the number of Majorana zero modes localized on the edge. There are three different phases classified by the total winding number and we adopt the terminology in the literature[14,15]:

(1) $\nu_\lambda = 0$, SSH-like trivial phase, no Majorana zero mode on the edge;

(2) $\nu_\lambda = 1$, Kitaev-like topological phase, one Majorana zero mode on the edge;

(3) $\nu_\lambda = 2$, SSH-like topological phase, two Majorana zero modes, i.e. one complex zero mode, on the edge.

The phase boundaries are $(t_y + \Delta_y)^2 = (t_x - \Delta_x)^2$ and $(t_y - \Delta_y)^2 = (t_x + \Delta_x)^2$. The energy spectrum of the

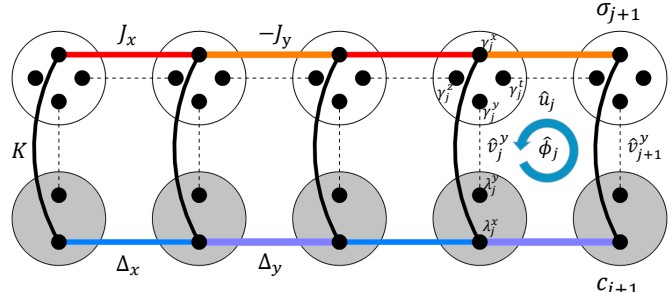

FIG. 4: The Majorana representations of the spin-charge ladder. Each white and grey circles denote one spin and one charge respectively, and each black point denotes one Majorana fermion. The horizontal dashed lines denote the $Z_2$ gauge field $\hat{u}_j$, and vertical dashed lines denote the $Z_2$ bosonic matter field $\hat{v}_j^y$. The circular arrow denotes the $Z_2$ Kondo flux $\hat{\phi}_j$ in a square plaquette. The black solid lines denote the Ising coupling, The spin-charge ladder is composed of coupled Majorana-SSH models.

superconducting wire are

$$\varepsilon_k^x = \pm\frac{1}{2}\sqrt{\left(t_x - \Delta_x + \left(t_y + \Delta_y\right)\cos k\right)^2 + \left(t_y + \Delta_y\right)^2 \sin^2 k},$$
(43)

$$\varepsilon_k^y = \pm\frac{1}{2}\sqrt{\left(t_x + \Delta_x + \left(t_y - \Delta_y\right)\cos k\right)^2 + \left(t_y - \Delta_y\right)^2 \sin^2 k},$$
(44)

where the gaps close at the phase boundaries.

## V. SPIN-CHARGE LADDER: COUPLED MAJORANA-SSH MODELS

The Ising-type interaction couples the spin chain and superconducting wire to form the spin-charge ladder. In the Majorana fermion representation, the Hamiltonian of the Ising coupling is

$$H_I = -\sum_j K i\gamma_j^x \lambda_j^x i\gamma_j^y \lambda_j^y = -\sum_j K \hat{v}_j^x \hat{v}_j^y, \qquad (45)$$

where the anticommutativity of $\gamma$ and $\lambda$ Majorana fermions is discussed in the Appendix A and

$$\hat{v}_j^\alpha = i\gamma_j^\alpha \lambda_j^\alpha. \qquad (46)$$

The operators $\hat{v}_j^\alpha$ ($\alpha = x, y$) satisfy $\left(\hat{v}_j^\alpha\right)^2 = 1$, and have $Z_2$ eigenvalues $\hat{v}_j^\alpha = \pm 1$. The spin-charge ladder can be viewed as coupled Majorana-SSH models as shown in FIG. 4. In the following, we'll derive the exact solution of the spin-charge ladder in the flat band limit, and study the $Z_2$ order fractionalization, topological phase transition, and odd frequency pairing therein.

## A. exact solution

The spin-charge ladder is exactly solvable when the parameters satisfy

$$t_x + \Delta_x = t_y - \Delta_y = 0, \qquad (47)$$

or

$$t_x - \Delta_x = t_y + \Delta_y = 0. \qquad (48)$$

According to Eq. (43) and (44), the above conditions correspond to one of the energy spectrum of the superconducting wire becomes complete flat. Without loss of generality, we focus on the flat band limit $t_x + \Delta_x = t_y - \Delta_y = 0$, and the exactly solvable Hamiltonian in the Majorana fermion representation is

$$H_E = \sum_j \left(J_x \hat{u}_{2j-1} i\gamma_{2j-1}^x \gamma_{2j}^x - J_y \hat{u}_{2j} i\gamma_{2j}^x \gamma_{2j+1}^x\right)$$
$$+ \sum_j \left(\Delta_x i\lambda_{2j-1}^x \lambda_{2j}^x + \Delta_y i\lambda_{2j}^x \lambda_{2j+1}^x\right)$$
$$- \sum_j K \hat{v}_j^y i\gamma_j^x \lambda_j^x. \qquad (49)$$

Under local $Z_2$ gauge transformation, the operators transform as

$$\gamma_j^x \to \Lambda_j \gamma_j^x, \qquad (50)$$
$$\hat{v}_j^y \to \Lambda_j \hat{v}_j^y, \qquad (51)$$
$$\hat{u}_j \to \Lambda_j \hat{u}_j \Lambda_{j+1}, \qquad (52)$$

the exactly solvable Hamiltonian $H_E$ has the local $Z_2$ gauge symmetry, and can be viewed as a lattice gauge theory that $\gamma_j^x$ and $\hat{v}_j^y$ are emergent $Z_2$ fermionic and bosonic matter fields with $Z_2$ gauge charges, and $\hat{u}_j$ are emergent $Z_2$ gauge field. Since

$$\left[H, \hat{u}_j\right] = 0, \qquad (53)$$
$$\left[H_E, \hat{v}_j^y\right] = 0, \qquad (54)$$

$\hat{u}_j$ and $\hat{v}_j^y$ are constants of motion in the flat band limit. As

$$\left[\hat{u}_j, \hat{u}_l\right] = 0, \qquad (55)$$
$$\left[\hat{v}_j^y, \hat{v}_l^y\right] = 0, \qquad (56)$$
$$\left[\hat{u}_j, \hat{v}_l^y\right] = 0, \qquad (57)$$

all $\hat{u}_j$ and $\hat{v}_j^y$ constitute a set of good quantum numbers. We can divide total Hilbert space into sectors $\{u, v^y\}$ characterized by the eigenvalues of all good quantum numbers. In each sector, the exactly solvable Hamiltonian describes free Majorana fermions coupled to static

$Z_2$ bosonic matter and gauge fields

$$H_E \left(\{u, v^y\}\right) = \sum_j \left(J_x u_{2j-1} i\gamma_{2j-1}^x \gamma_{2j}^x - J_y u_{2j} i\gamma_{2j}^x \gamma_{2j+1}^x\right)$$
$$+ \sum_j \left(\Delta_x i\lambda_{2j-1}^x \lambda_{2j}^x + \Delta_y i\lambda_{2j}^x \lambda_{2j+1}^x\right)$$
$$- \sum_j K v_j^y i\gamma_j^x \lambda_j^x, \tag{58}$$

and such quadratic Hamiltonian is exactly solvable. In each sector, the ground state is obtained by diagonalizing the quadratic Hamiltonian $H_E \left(\{u, v^y\}\right)$ and filling all negative energy levels. The ground state energy $E_0 \left(\{u, v^y\}\right)$ is a functional of $\{u, v^y\}$. The ground state sector is determined by minimizing $E_0 \left(\{u, v^y\}\right)$.

To determine the ground state sector, we first consider the strong coupling limit $|K| \gg |J_{x,y}|, |\Delta_{x,y}|$. In term of the exactly solvable Hamiltonian, there are four local states

$$\left| v_j^y = \pm 1, i\gamma_j^x \lambda_j^x = \pm 1 \right\rangle. \tag{59}$$

In the strong coupling limit, the Ising coupling splits four local states into two high-energy states

$$\left| v_j^y \mathrm{sgn}\left(K\right) = -i\gamma_j^x \lambda_j^x = \pm 1 \right\rangle, \tag{60}$$

and two low-energy states

$$\left| v_j^y \mathrm{sgn}\left(K\right) = i\gamma_j^x \lambda_j^x = \pm 1 \right\rangle. \tag{61}$$

We can derive the effective Hamiltonian in the $2^N$-dimensional low-energy subspace by treating the spin chain and superconducting wire as perturbations, where $N$ is the number of total sites. To the leading order, the effective Hamiltonian is given by

$$H_{\mathrm{eff}} = \frac{1}{2\,|K|} \sum_j \left(\Delta_x J_x \hat{\phi}_{2j-1} - \Delta_y J_y \hat{\phi}_{2j}\right), \tag{62}$$

where

$$\hat{\phi}_j = \hat{v}_j^y \hat{u}_j \hat{v}_{j+1}^y \tag{63}$$

is a gauge invariant $Z_2$ Kondo flux operator[7]. By the inverse Jordan-Wigner transformation, the $Z_2$ Kondo flux operators in terms of spin and charge degrees of freedom are given by

$$\hat{\phi}_{2j-1} = -\sigma_{2j-1}^y \sigma_{2j}^y i\lambda_{2j-1}^y \lambda_{2j}^y, \tag{64}$$
$$\hat{\phi}_{2j} = \sigma_{2j}^x \sigma_{2j+1}^x i\lambda_{2j}^y \lambda_{2j+1}^y, \tag{65}$$

where two spin operators and two charge operators form a Kondo flux in a square plaquette as shown in FIG. 4. By definition, the operators $\hat{\phi}_j$ have $Z_2$ eigenvalues $\phi_j = \pm 1$, and are constants of motion in the flat band limit. The form of the effective Hamiltonian manifests the ground state energy is actually a functional of gauge invariant

quantities $\{\phi\}$ as it must be. Thus the ground state sector is determined by minimizing $E_0(\{\phi\})$. In the strong coupling limit, the ground state sector is determined by the relations

$$\mathrm{sgn}\left(\Delta_x J_x\right)\hat{\phi}_{2j-1} = -1, \tag{66}$$
$$-\mathrm{sgn}\left(\Delta_y J_y\right)\hat{\phi}_{2j} = -1, \tag{67}$$

which results are consistent with the Lieb's theorem[16]. By absorbing the signs of the parameters into the flux operators as defined by Lieb, the ground state sector of the flux phase of half-filled band is $\pi$-flux for each square plaquette.

For generic parameters, we have to resort to numerical calculations to find the ground state sector. As the numbers of $u_j$ and $v_j^y$ are both $N$, we need to traverse all the $2^{2N}$ sectors and find the sector with lowest ground state energy. The numerical results on small lattice size not only confirm the results in the strong coupling limit, but also indicate that the relations are valid for generic parameters. As a physical quantity, the ground state energy only depends on the gauge invariant quantities $\{\phi\}$. The number of different choices of $u_j$ and $v_j^y$ corresponding to each flux sector $\{\phi\}$ is $2^{N+1}$, where $N$ comes from the number of local $Z_2$ gauge transformations and 1 comes from the global $Z_2$ transformation $v_j^y \to -v_j^y$. Therefore, each state is $2^{N+1}$-fold degenerate.

## B. $Z_2$ order fractionalization

The global $Z_2$ transformation related two-fold degeneracy of the ground states indicates the global $Z_2$ symmetry breaking of the spin-charge ladder. The $Z_2$ bosonic matter field $\hat{v}_j^\alpha$ can serve as the order parameter. In the Kitaev-Kondo model, the $SU\left(2\right)$ spin symmetric Kondo coupling allows the Majorana spinon and electron to form a bound state, and the composite bosonic spinor inherits the fractional quantum numbers of spinon and electron. The condensation of the spin $\frac{1}{2}$ and charge $e$ bosonic spinor renders into the fractionalized order[7]. The Ising coupling of the spin-charge ladder in Eq. (45) tends to form the bound state of Majorana spinon and charge, and the composite bosonic scalar inherits the fractional quantum numbers spin $\frac{1}{2}$ and $Z_2$ charge. The condensation of the bosonic scalar renders into $Z_2$ order fractionalization. From the inverse Jordan-Wigner transformation, the order parameters of the $Z_2$ order fractionalization in terms of spin degrees of freedom are essentially string

order parameters

$$\hat{v}^x_{2j-1} = i\gamma^x_{2j-1}\lambda^x_{2j-1}$$
$$= i\sigma^y_{2j-1}\big(\prod_{l<2j-1}\sigma^z_l\big)\lambda^x_{2j-1}, \tag{68}$$

$$\hat{v}^y_{2j-1} = i\gamma^y_{2j-1}\lambda^y_{2j-1}$$
$$= i\sigma^x_{2j-1}\big(\prod_{l<2j-1}\sigma^z_l\big)\lambda^y_{2j-1}, \tag{69}$$

$$\hat{v}^x_{2j} = i\gamma^x_{2j}\lambda^x_{2j}$$
$$= -i\sigma^x_{2j}\big(\prod_{l<2j}\sigma^z_l\big)\lambda^x_{2j}, \tag{70}$$

$$\hat{v}^y_{2j} = i\gamma^y_{2j}\lambda^y_{2j}$$
$$= i\sigma^y_{2j}\big(\prod_{l<2j}\sigma^z_l\big)\lambda^y_{2j}. \tag{71}$$

The order parameters transform nontrivially under the $Z_2$ total spin parity

$$Z^s_2 = \prod_j \sigma^z_j = \prod_j i\gamma^x_j\gamma^y_j, \tag{72}$$

and the $Z_2$ total charge number parity

$$Z^c_2 = (-1)^{\sum_j n_j} = \prod_j (-i\lambda^x_j\lambda^y_j), \tag{73}$$

as follows

$$Z^s_2\hat{v}^\alpha_j Z^s_2 = -\hat{v}^\alpha_j, \tag{74}$$
$$Z^c_2\hat{v}^\alpha_j Z^c_2 = -\hat{v}^\alpha_j. \tag{75}$$

The spin parity is defined as even (odd) for spin up (down). The condensation of the bosonic scalar breaks both the $Z_2$ total spin and charge number parity symmetries. In particular, the exactly solvable Hamiltonian Eq. (49) in the axial gauge has the spin-charge duality

$$\gamma^x_j \leftrightarrow \lambda^x_j, \tag{76}$$
$$\gamma^y_j \leftrightarrow -\lambda^y_j, \tag{77}$$
$$J_x \leftrightarrow \Delta_x, \tag{78}$$
$$J_y \leftrightarrow -\Delta_y, \tag{79}$$

and the two $Z_2$ parity symmetries interchange

$$Z^s_2 \leftrightarrow Z^c_2. \tag{80}$$

Alternatively, we can introduce the other equivalent order parameters

$$\hat{O}_{j,\pm} = \frac{1}{2}(\hat{v}^x_j \pm \hat{v}^y_j), \tag{81}$$

and the Ising coupling can be rewritten as

$$H_I = -\sum_j K\left(2\hat{O}^2_{j,+} - 1\right)$$
$$= -\sum_j K\left(1 - 2\hat{O}^2_{j,-}\right)$$
$$= -\sum_j K\left(\hat{O}^2_{j,+} - \hat{O}^2_{j,-}\right), \tag{82}$$

which form indicates positive $K$ favors $\hat{O}_{j,+}$ while negative $K$ favors $\hat{O}_{j,-}$ to lower the energy.

We can calculate the expectation values of the order parameters analytically with the help of the exact solution. We define the complex $f$-fermions

$$f_{2j-1} = \frac{1}{2}\left(\lambda^x_{2j-1} + i\gamma^x_{2j-1}\right), \tag{83}$$
$$f_{2j} = \frac{1}{2}\left(\gamma^x_{2j} - i\lambda^x_{2j}\right), \tag{84}$$

and the exactly solvable Hamiltonian in the complex $f$-fermion representation is

$$H_E = \sum_j\big[\left(-\Delta_x - J_x\hat{u}_{2j-1}\right)f^\dagger_{2j-1}f_{2j} + \left(\Delta_y - J_y\hat{u}_{2j-1}\right)f^\dagger_{2j}f_{2j+1} + h.c.\big]$$
$$+ \sum_j\big[\left(\Delta_x - J_x\hat{u}_{2j-1}\right)f^\dagger_{2j-1}f^\dagger_{2j} + \left(\Delta_y + J_y\hat{u}_{2j-1}\right)f^\dagger_{2j}f^\dagger_{2j+1} + h.c.\big]$$
$$+ \sum_j K\hat{v}^y_j\big(f^\dagger_jf_j - f_jf^\dagger_j\big). \tag{85}$$

In the ground state sectors, $Z_2$ Kondo fluxes are uniform

$$\hat{\phi}_{2j-1} = \phi_A, \tag{86}$$
$$\hat{\phi}_{2j} = \phi_B, \tag{87}$$

and satisfy the relations

$$\text{sgn}\left(\Delta_x J_x\right)\phi_A = -1, \tag{88}$$
$$-\text{sgn}\left(\Delta_y J_y\right)\phi_B = -1. \tag{89}$$

We take the gauge

$$\hat{v}_{2j-1}^y = v_A = v, \tag{90}$$
$$\hat{v}_{2j}^y = v_B = v, \tag{91}$$

then in the ground state sectors we have

$$\hat{u}_{2j-1} = u_A = \phi_A, \tag{92}$$
$$\hat{u}_{2j} = u_B = \phi_B. \tag{93}$$

We perform the Fourier transformation of complex $f$-fermion operators

$$f_{r\mu} = \frac{1}{\sqrt{N_c}} \sum_k e^{ikr} f_{k\mu}, \tag{94}$$

the exactly solvable Hamiltonian in momentum space is

$$H_E(\{\phi_A, \phi_B\}) = \sum_k \Phi_k^\dagger \hat{h}_E(k) \Phi_k, \tag{95}$$

$$\hat{h}_E(k) = \begin{pmatrix} Kv & z_k & 0 & w_k \\ z_k^* & Kv & -w_k^* & 0 \\ 0 & -w_k & -Kv & -z_k \\ w_k^* & 0 & -z_k^* & -Kv \end{pmatrix}, \tag{96}$$

where the four-component vector of complex $f$-fermion is $\Phi_k = \begin{pmatrix} f_{kA} & f_{kB} & f_{-kA}^\dagger & f_{-kB}^\dagger \end{pmatrix}^T$, and the matrix elements of $\hat{h}_E(k)$ are

$$z_k = \frac{1}{2}\left[(-\Delta_x - J_x u_A) + (\Delta_y - J_y u_B)e^{-ik}\right], \tag{97}$$

$$w_k = \frac{1}{2}\left[(\Delta_x - J_x u_A) - (\Delta_y + J_y u_B)e^{-ik}\right]. \tag{98}$$

To obtain concise analytical results, we consider the limit

$$J_x = -J_y = J, \tag{99}$$
$$\Delta_x = \Delta_y = \Delta, \tag{100}$$

and expectation values of the order parameters are given by

$$\begin{aligned} O_\pm &= \langle \hat{O}_{j,\pm} \rangle \\ &= \frac{v}{2}\left(\frac{1}{N_c}\sum_k \frac{K}{\sqrt{(Ju-\Delta)^2 \sin^2 k + K^2}} \pm 1\right), \end{aligned} \tag{101}$$

where

$$u_A = u_B = u = -\text{sgn}(J\Delta), \tag{102}$$

and the calculation details are given in the Appendix B. From the analytical expression Eq. (101), the two order parameters both have nonzero expectation values and coexist with each other as they break both the $Z_2$ total spin and charge number parity symmetries. Meanwhile, the magnitudes of two order parameters compete with each other as positive (negative) $K$ favors $\hat{O}_{j,+}$ ($\hat{O}_{j,-}$). The intertwined order parameters coexist and compete with

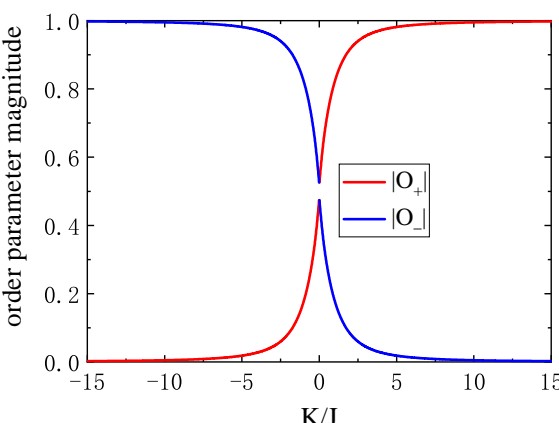

FIG. 5: (Color online) The magnitudes of order parameters $O_+$ (red) and $O_-$ (blue) vs. $K/J$. The parameters are $-J_x = J_y = \Delta_x = \Delta_y = -J$. The singular point at $K = 0$ is due to the macroscopic degeneracy. The two order parameters coexist and compete with each other.

each other in the spin-charge ladder as shown in FIG. 5 may provide essential insight into the $Z_2$ order fractionalization. The singular point at $K = 0$ in FIG. 5 is due to the $2^N$-fold degeneracy, i.e. macroscopic degeneracy, caused by the absence of $N$ operators $\hat{v}_j^y$'s in the exact solvable Hamiltonian. We emphasize that the coexistence of two order parameters must take count of the fluctuations beyond the mean-field theory as different forms of Ising coupling in Eq. (82) imply the mean-field theory tends to one nonzero order parameter to lower the energy. Certainly the exact solution has taken count of all fluctuations.

From the perspective of the lattice gauge theory, the $Z_2$ order fractionalization can be viewed as the $Z_2$ Higgs phase in the sense that $Z_2$ bosonic matter field has spontaneous $Z_2$ symmetry breaking and $Z_2$ gauge field is massive[17]. However, the exactly solvable Hamiltonian in Eq. (49) indicates the bosonic matter field and gauge field are coupled by integrating the fermionic matter field. Therefore the fermionic excitations are also one of characteristics of the $Z_2$ order fractionalization.

### C. topological phase transition

We shall study the low-energy fermionic excitations and the quantum phase transitions in the spin-charge ladder. By diagonalizing the matrix $\hat{h}_E(k)$, the energy spectrum of fermionic excitations in the ground state sectors is

$$\left(E_k^\pm\right)^2 = K^2 + |z_k|^2 + |w_k|^2 \pm \sqrt{|2Kz_k|^2 + (z_k w_k^* + z_k^* w_k)^2}. \tag{103}$$

At $k = 0, \pi$, we have $z_k = z_k^*$, $w_k = w_k^*$, and the energy spectrum simplifies to

$$\left(E_{k=0,\pi}^{\pm}\right)^2 = \left(\sqrt{K^2 + w_{k=0,\pi}^2} \pm \sqrt{z_{k=0,\pi}^2}\right)^2. \quad (104)$$

$E_k^+$ is always gapped, and the gapless conditions of $E_k^-$ are

$$K^2 + w_{k=0,\pi}^2 = z_{k=0,\pi}^2, \quad (105)$$

or equivalently

$$\left(K^2 - \Delta_x J_x u_A + \Delta_y J_y u_B\right)^2 = \left(\Delta_x J_y u_B - \Delta_y J_x u_A\right)^2, \quad (106)$$

which indicate quantum phase transitions at the gapless points.

To explore the nature of the quantum phase transitions, we focus on the low-energy physics of the spin-charge ladder, which is captured by the exactly solvable Hamiltonian in the ground state sectors $H_E\left(\{\phi_A, \phi_B\}\right)$ in Eq. (95). As the Ising coupling effectively acts as a chemical potential, the Hamiltonian $H_E\left(\{\phi_A, \phi_B\}\right)$ breaks the sublattice chiral symmetry

$$\tau^z \hat{h}_E\left(k\right) \tau^z \neq -\hat{h}_E\left(k\right), \quad (107)$$

nonetheless the Hamiltonian $H_E\left(\{\phi_A, \phi_B\}\right)$ has another particle-hole chiral symmetry

$$\rho^x \hat{h}_E\left(k\right) \rho^x = -\hat{h}_E\left(k\right), \quad (108)$$

where $\rho^\alpha (\alpha = x, y, z)$ are Pauli matrices acting on the particle-hole degrees of freedom, and the above symmetries are defined in terms of complex $f$-fermions. With particle-hole chiral symmetry, we can define the spectral chiral index of complex $f$-fermions in the ground states as a topological invariant of the spin-charge ladder

$$C_E = \frac{1}{4\pi i} \mathrm{Tr} \int_{-\pi}^{\pi} dk \rho^x \hat{g}_E^{-1}\left(k\right) \partial_k \hat{g}_E\left(k\right), \quad (109)$$

where $\hat{g}_E\left(k\right) = -\hat{h}_E^{-1}\left(k\right)$ is the Green's function at zero frequency[18–21]. The spectral chiral index $C_E$ is given by

$$\left|C_E\right| = \Theta\left[\left(\Delta_x J_y u_B - \Delta_y J_x u_A\right)^2 - \left(K^2 - \Delta_x J_x u_A + \Delta_y J_y u_B\right)^2\right] \quad (110)$$

and the calculation details are given in the Appendix C. There are two phases classified by the absolute value of the spectral chiral index:

(1) $|C_E| = 0$, SSH-like phase, no robust Majorana zero mode on the edge;

(2) $|C_E| = 1$, Kitaev-like phase, one robust Majorana zero mode on the edge.

The phase boundaries determined by the spectral chiral index in Eq. (110) are consistent with the gapless conditions of the energy spectrum in Eq. (106). The absolute value of the spectral chiral index counts the number of robust Majorana zero modes localized on the edge. In the

FIG. 6: Dyson's equation of the propagator $G_C$ (thick solid line) of the charge degrees of freedom. $G_C^0$ (thin solid line) and $G_S^0$ (thin dashed line) are the bare propagators of the superconducting wire and spin chain respectively, and $V_k$ is the vertex hybridizing charge and spin degrees of freedom due to Ising coupling.

Majorana-SSH model, the Majorana zero modes are *fragile* in the sense of protected by the sublattice chiral symmetry, and away from half-filling the chemical potential term will break the sublattice chiral symmetry and couple two Majorana-SSH models then split the Majorana zero modes. However, as shown in FIG. 4, the low-energy physics of the spin-charge ladder in the flat-band limit is captured by the dimerized Kitaev-chain. Therefore the Majorana zero modes are *robust* in the Kitaev-like phase, just as in the Kitaev-chain.

The topological phase transition of the spin-charge ladder has a physical picture in real space in the perspective of coupled Majorana-SSH models. In the ground state sectors, we have the relations Eq. (88) and (92), and the gapless conditions simplify to

$$K^2 = \left(|\Delta_x| - |\Delta_y|\right)\left(|J_y| - |J_x|\right). \quad (111)$$

In the weak coupling limit $|K| \ll |J_{x,y}|, |\Delta_{x,y}|$, the conditions for the Kitaev-like topological superconductor (TSC) phase are $|\Delta_x| \lessgtr |\Delta_y|$ and $|J_y| \lessgtr |J_x|$, which means one of the coupled Majorana-SSH models is in the topological phase, meanwhile another is in the trivial phase, and Majorana fermions tend to pair on the strong bonds within their own Majorana-SSH models, i.e. the horizontal solid lines in FIG. 4. Eventually one Majorana zero mode is left unpaired on the edge. In the strong coupling limit $|K| \gg |J_{x,y}|, |\Delta_{x,y}|$, Majorana fermions tend to pair between the Majorana-SSH models, i.e. the vertical solid lines in FIG. 4, and no Majorana zero mode is left unpaired on the edge. However we emphasize the Kitaev-like TSC phase is not adiabatically connected to the topological phase of the non-interacting Kitaev chain. Not only the low-energy physics contains Majorana fermions from spin-chain and superconducting wire, but also there exist gapped excitations of $Z_2$ Kondo flux. The $Z_2$ Kondo flux excitation is similiar to the vison in the $Z_2$ gauge theory of electron fractionalization[5], and the Kitaev-like TSC phase is denoted as TSC*.

### D.  odd frequency pairing

We shall investigate the effect of the spin chain on the pairing properties of the superconducting wire due to the Ising coupling. We focus on the flat band limit in

Eq. (47), and the momentum space Hamiltonian of the superconducting wire in the complex $c$-fermion representation is

$$H_C = \sum_k \Psi_k^\dagger \hat{h}_C(k) \Psi_k, \qquad (112)$$

$$\hat{h}_C(k) = \begin{pmatrix} 0 & g_k & 0 & -g_k \\ g_k^* & 0 & g_k^* & 0 \\ 0 & g_k & 0 & -g_k \\ -g_k^* & 0 & -g_k^* & 0 \end{pmatrix}, \qquad (113)$$

where the four-component vector of complex $c$-fermion is $\Psi_k = \begin{pmatrix} c_{kA} & c_{kB} & c_{-kA}^\dagger & c_{-kB}^\dagger \end{pmatrix}^T$, and the matrix elements of $\hat{h}_C(k)$ are

$$g_k = \frac{1}{2}\left(-\Delta_x + \Delta_y e^{-ik}\right). \qquad (114)$$

The inverse bare propagator of the superconducting wire is

$$
\begin{aligned}
&\left(G_C^0\right)^{-1}(k,\omega) \\
&= \omega - \hat{h}_C(k) \\
&= \omega - \rho^z\left(g_k^x \tau^x + g_k^y \tau^y\right) - \rho^y\left(-g_k^y \tau^x + g_k^x \tau^y\right), \quad (115)
\end{aligned}
$$

where terms proportional to $\rho^z$ and $\rho^y$ come from hopping and pairing parts respectively, and

$$g_k^x = \frac{1}{2}\left(-\Delta_x + \Delta_y \cos k\right), \qquad (116)$$

$$g_k^y = \frac{1}{2}\Delta_y \sin k. \qquad (117)$$

Similarly, the inverse bare propagator of the spin chain is

$$
\begin{aligned}
\left(G_S^0\right)^{-1}(k,\omega) &= \omega - \hat{h}_S(k) \\
&= \omega - d_k^x \tau^x - d_k^y \tau^y. \qquad (118)
\end{aligned}
$$

We focus on the low-energy physics of the spin-charge ladder and work within the ground state sectors. The exactly solvable Hamiltonian in the ground state sectors can be written as

$$H_E = \sum_k \begin{pmatrix} \Psi_k^\dagger & \Gamma_k^\dagger \end{pmatrix} \begin{pmatrix} \hat{h}_C(k) & V_k \\ V_k^\dagger & \hat{h}_S(k) \end{pmatrix} \begin{pmatrix} \Psi_k \\ \Gamma_k \end{pmatrix}, \quad (119)$$

where $V_k$ is the hybridization matrix between the four-component vector $\Psi_k$ and two-component vector $\Gamma_k$ due to Ising coupling

$$V_k = \frac{iK}{\sqrt{2}} \begin{pmatrix} v_A & 0 \\ 0 & -v_B \\ v_A & 0 \\ 0 & v_B \end{pmatrix}. \qquad (120)$$

For the spin-charge coupled system, we can integrate the spin degrees of freedom and obtain the effective action of the charge degrees of freedom, or equivalently we can

obtain the propagator of the charge degrees of freedom from the Dyson's equation as shown in FIG. 6

$$G_C = G_C^0 + G_C^0 \Sigma_C G_C, \qquad (121)$$

where $\Sigma_C$ is the self-energy and can be divided into normal and pairing parts

$$
\begin{aligned}
\Sigma_C(k,\omega) &= V_k G_S^0(k,\omega) V_k^\dagger \\
&= \Sigma_C^N(k,\omega) + \Sigma_C^{SC}(k,\omega). \qquad (122)
\end{aligned}
$$

The straightforward calculations give

$$\Sigma_C^N(k,\omega) = \frac{K^2}{2} \frac{\omega - v_A v_B \rho^z\left(d_k^x \tau^x + d_k^y \tau^y\right)}{\omega^2 - \left(d_k^x\right)^2 - \left(d_k^y\right)^2}, \qquad (123)$$

$$
\begin{aligned}
\Sigma_C^{SC}(k,\omega) = &-\frac{K^2}{2} \frac{v_A v_B \rho^y\left(-d_k^y \tau^x + d_k^x \tau^y\right)}{\omega^2 - \left(d_k^x\right)^2 - \left(d_k^y\right)^2} \\
&+ \frac{K^2}{2} \frac{\omega \rho^x \tau^z}{\omega^2 - \left(d_k^x\right)^2 - \left(d_k^y\right)^2}. \qquad (124)
\end{aligned}
$$

From the Dyson's equation

$$G_C^{-1} = \left(G_C^0\right)^{-1} - \Sigma_C, \qquad (125)$$

and compare with the inverse bare propagator in Eq. (115), besides some corrections to the hopping and pairing parts, the Ising coupling generates a new term

$$
\begin{aligned}
\Sigma_C^{\text{odd}}(k,\omega) &= -\Sigma_C^{\text{odd}}(k,-\omega) \\
&= \frac{K^2}{2} \frac{\omega \rho^x \tau^z}{\omega^2 - \left(d_k^x\right)^2 - \left(d_k^y\right)^2}, \qquad (126)
\end{aligned}
$$

which describes odd frequency, even parity, onsite pairing. A careful look at the origin of the odd frequency pairing, the essential ingredient is the fermionic propagator $G_S^0$ of the spin chain. Such an odd frequency pairing can't be generated in electron-phonon or electron-magnon coupled systems as the propagators of phonon or magnon are bosonic and even in frequency. The Majorana fractionalization of the spins in spin-electron coupled systems provides a general mechanism for odd frequency pairing[22–25].

## VI. DISCUSSION

In summary, we study a spin-charge ladder model and discover the exact solution at the flat band limit. With the help of the exact solution, we explore the $Z_2$ order fractionalization with bosonic scalar order parameters. The order parameters compose of Majorana spinon and charge with fractional quantum numbers spin $\frac{1}{2}$ and $Z_2$ charge, and are essentially string order parameters. In the $Z_2$ order fractionalized phase, two dual $Z_2$ symmetries break spontaneously that lead to the two intertwined order parameters coexisting and competing with

each other. The low-energy fermionic excitations in the $Z_2$ order fractionalized phase are gapped except at the critical points, where the spin-charge ladder undergoes a topological phase transition characterized by the spectral chiral index. The topological phase is denoted as TSC* to emphasize the correlated nature with gapped $Z_2$ Kondo flux excitation. The Majorana fractionalization of the spins, i.e. Majorana spinons in the spin chain, effectively generates odd frequency pairing in the superconducting wire. The exact solution describes a lattice gauge theory that $Z_2$ fermionic and bosonic matter fields coupled with $Z_2$ gauge field, and $Z_2$ order fractionalization is a $Z_2$ Higgs phase that $Z_2$ bosonic matter field has spontaneous $Z_2$ symmetry breaking and $Z_2$ gauge field is massive. Combined with the pioneering works[7,9], such a concrete and exact study lays the foundation for the future research about order fractionalization.

In the main text, we focus on the exact solution and derive exact results about order fractionalization in the spin-charge ladder. Away from the flat band limit, $\lambda^y$ Majorana fermions become itinerant, and operators $\hat{v}_j^y$ are no longer constants of motion and $Z_2$ bosonic matter field acquires fluctuations and becomes dynamic. With increasing fluctuations, a phase transition belonging to the $1+1$-D Ising universality class is presumed as the $Z_2$ characteristic of the fluctuating bosonic scalar. More interestingly, the Ising coupling is tuned close to the critical points and the low-energy fermionic exciations are Dirac fermions, then along the critical points, the coupling of fluctuating bosonic scalar and Dirac fermions may generate novel quantum criticality[26]. The state-of-the-art DMRG provides a powerful method to the future systematic study of the whole phase diagram of the spin-charge ladder.

The order fractionalization provides a new mechanism of unconventional superconductivity. The key is the Majorana fractionalization of spins. For electron-spin coupled systems, spins form the intermediary spin liquid state with Majornana spinons. Then the Kondo coupling leads to the order fractionalization that breaks electron's global $U(1)$ symmetry. The electron superconductivity inherits from Majorana spinons[27], such as the topological superconductivity[28], Odd-frequency pair density wave[29], and spin-triplet pairing in the Kitaev-Kondo model[7]. As the amplitude of the effective odd frequency pairing is hugely enhanced with zero energy bound state and is inversely proportional to $\omega$[30–32], the TSC* phase in the spin-charge ladder with Majorana zero mode is possibly detected in a Josephson junction setup[33,34].

### Acknowledgment

This work was supported by the National Key R&D Program of China (Grants No. 2022YFA1403700), NSFC (Grants No. 12141402, 12334002), the Science, Technology and Innovation Commission of Shenzhen Municipality (No. ZDSYS20190902092905285), the SUSTech-NUS Joint Research Program, and Center for Computational Science and Engineering at Southern University of Science and Technology.

## Appendix A: Anticommutativity of $\gamma$ and $\lambda$ Majorana fermions

It is presumed that the $\gamma$ Majorana fermions of the spin chain and the $\lambda$ Majorana fermions of the superconducting wire anticommute. The inverse Jordan-Wigner transformation indicates $\gamma$ Majorana fermions are spin string operators

$$\gamma_{2j-1}^x = \sigma_{2j-1}^y \prod_{l<2j-1} \sigma_l^z, \tag{A1}$$

$$\gamma_{2j-1}^y = \sigma_{2j-1}^x \prod_{l<2j-1} \sigma_l^z, \tag{A2}$$

$$\gamma_{2j}^x = -\sigma_{2j}^x \prod_{l<2j} \sigma_l^z, \tag{A3}$$

$$\gamma_{2j}^y = \sigma_{2j}^y \prod_{l<2j} \sigma_l^z, \tag{A4}$$

and the spin operators certainly commute with the $\lambda$ Majorana fermions

$$\left[\sigma_j^\alpha, \lambda_l^\beta\right] = 0, \tag{A5}$$

thus $\gamma$ and $\lambda$ Majorana fermions actually commute with each other

$$\left[\gamma_j^\alpha, \lambda_l^\beta\right] = 0. \tag{A6}$$

In consideration of spin and charge degrees of freedom are coupled, we introduce a new set of $\tilde{\lambda}$ Majorana fermions

$$\tilde{\lambda}_j^\alpha = \lambda_j^\alpha Z_2^s, \tag{A7}$$

where

$$Z_2^s = \prod_j \sigma_j^z = \prod_j i\gamma_j^x \gamma_j^y, \tag{A8}$$

is the spin parity. As

$$\left\{\gamma_j^\alpha, Z_2^s\right\} = 0, \tag{A9}$$

$\gamma$ and $\tilde{\lambda}$ Majorana fermions anticommute with each other

$$\left\{\gamma_j^\alpha, \tilde{\lambda}_l^\beta\right\} = 0. \tag{A10}$$

In the main text, we still use the notation $\lambda$ instead of $\tilde{\lambda}$ for simplicity, and assume $\gamma$ and $\lambda$ Majorana fermions anticommute with each other if no ambiguities.

## Appendix B: Expectation values of order parameters

We shall calculate the expectation values of order parameters in the Majorana fermion representation. In the limit

$$J_x = -J_y = J, \tag{B1}$$
$$\Delta_x = \Delta_y = \Delta, \tag{B2}$$

the exactly solvable Hamiltonian in the ground state sectors is

$$H_E\left(\{\phi\}\right) = \sum_j \left(Jui\gamma_j^x\gamma_{j+1}^x + \Delta i\lambda_j^x\lambda_{j+1}^x - Kvi\gamma_j^x\lambda_j^x\right), \tag{B3}$$

We perform the Fourier transformation

$$\gamma_j^x = \sqrt{\frac{2}{N}}\sum_q e^{iqj}\gamma_q^x, \tag{B4}$$

$$\lambda_j^x = \sqrt{\frac{2}{N}}\sum_q e^{iqj}\lambda_q^x, \tag{B5}$$

and the exactly solvable Hamiltonian in momentum space is

$$H_E\left(\{\phi\}\right)$$
$$= \sum_q \left(\begin{array}{cc}\gamma_{-q}^x & \lambda_{-q}^x\end{array}\right)\left(\begin{array}{cc}-2Ju\sin q & -Kvi \\ Kvi & -2\Delta\sin q\end{array}\right)\left(\begin{array}{c}\gamma_q^x \\ \lambda_q^x\end{array}\right), \tag{B6}$$

To diagonalize the exactly solvable Hamiltonian, we perform the unitary transformation

$$\left(\begin{array}{c}\gamma_q^x \\ \lambda_q^x\end{array}\right) = \left(\begin{array}{cc}u_q^* & v_q \\ -v_q^* & u_q\end{array}\right)\left(\begin{array}{c}\eta_q^+ \\ \eta_q^-\end{array}\right), \tag{B7}$$

where

$$u_q = \frac{h_q + h_q^z}{\sqrt{2h_q\left(h_q + h_q^z\right)}} = u_q^*, \tag{B8}$$

$$v_q = \frac{ih_q^y}{\sqrt{2h_q\left(h_q + h_q^z\right)}} = -v_q^*, \tag{B9}$$

and

$$h_q^y = Kv, \tag{B10}$$
$$h_q^z = -\left(Ju - \Delta\right)\sin k, \tag{B11}$$
$$h_q = \sqrt{\left(Ju - \Delta\right)^2\sin^2 k + K^2}, \tag{B12}$$

The expectation value of $\hat{v}_j^x$ is

$$\langle\hat{v}_j^x\rangle = \frac{2}{N}\sum_q -iv_q u_q = \frac{1}{N}\sum_q \frac{Kv}{h_q}, \tag{B13}$$

and the expectation values of order parameters are

$$O_\pm = \langle\hat{O}_{j,\pm}\rangle$$
$$= \frac{1}{2}\left(\langle\hat{v}_{j,x}\rangle \pm \langle\hat{v}_{j,y}\rangle\right)$$
$$= \frac{v}{2}\left(\frac{1}{N}\sum_q \frac{K}{h_q} \pm 1\right). \tag{B14}$$

## Appendix C: Spectral chiral index

We shall calculate the spectral chiral index of the complex $f$-fermions. To work in the diagonal basis of particle-hole chiral symmetry operator, we perform a unitary transformation

$$U\rho^x U^\dagger = \rho^z, \tag{C1}$$

$$U\hat{h}_E\left(k\right)U^\dagger = \left(\begin{array}{cc}0 & V_E\left(k\right) \\ V_E^\dagger\left(k\right) & 0\end{array}\right) \tag{C2}$$

where

$$U = \frac{1}{\sqrt{2}}\left(\begin{array}{cccc}1 & 0 & 1 & 0 \\ 0 & 1 & 0 & 1 \\ -i & 0 & i & 0 \\ 0 & -i & 0 & i\end{array}\right), \tag{C3}$$

and

$$V_E\left(k\right) = \left(\begin{array}{cc}iKv & i\left(z_k - w_k\right) \\ i\left(z_k^* + w_k^*\right) & iKv\end{array}\right). \tag{C4}$$

The spectral chiral index is

$$C_E = \frac{1}{4\pi i}\mathrm{Tr}\int_{-\pi}^{\pi}dk\left(\begin{array}{cc}1 & 0 \\ 0 & -1\end{array}\right)\left(\begin{array}{cc}0 & -V_E\left(k\right) \\ -V_E^\dagger\left(k\right) & 0\end{array}\right)$$
$$\times\partial_k\left(\begin{array}{cc}0 & -V_E^{\dagger-1}\left(k\right) \\ -V_E^{-1}\left(k\right) & 0\end{array}\right)$$
$$= \frac{1}{4\pi i}\mathrm{Tr}\int_{-\pi}^{\pi}dk\left(V_E^{\dagger-1}\left(k\right)\partial_k V_E^\dagger\left(k\right) - V_E^{-1}\left(k\right)\partial_k V_E\left(k\right)\right)$$
$$= -\frac{1}{2\pi i}\mathrm{Tr}\int_{-\pi}^{\pi}dk\partial_k\ln V_E\left(k\right)$$
$$= -\frac{1}{2\pi i}\int_{-\pi}^{\pi}dk\partial_k\ln\det V_E\left(k\right), \tag{C5}$$

where

$$\det V_E\left(k\right) = -K^2 + \left(z_k - w_k\right)\left(z_k^* + w_k^*\right)$$
$$= -K^2 + \left(\Delta_x J_x u_A - \Delta_y J_y u_B\right)$$
$$+ \left(\Delta_x J_y u_B - \Delta_y J_x u_A\right)\cos k$$
$$+ i\left(\Delta_x J_y u_B + \Delta_y J_x u_A\right)\sin k, \tag{C6}$$

The spectral chiral index $C_E$ is the winding number of $\det V_E\left(k\right)$, and is determined by the cross points of the real axis at $k = 0$ and $k = \pi$

$$|C_E| = \begin{cases}1, & \det V_E\left(0\right)\det V_E\left(\pi\right) < 0 \\ 0, & \det V_E\left(0\right)\det V_E\left(\pi\right) > 0\end{cases} \tag{C7}$$

Note the spectral chiral index depends on the signs of the pairing potentials, which is a matter of global phase choice. As

$$\det V_E\left(0\right)\det V_E\left(\pi\right)$$
$$=\left(K^2-\Delta_x J_x u_A+\Delta_y J_y u_B\right)^2-\left(\Delta_x J_y u_B-\Delta_y J_x u_A\right)^2, \quad \text{(C8)}$$

the spectral chiral index is given by

$$\left|C_E\right|=\Theta\left[\left(\Delta_x J_y u_B-\Delta_y J_x u_A\right)^2-\left(K^2-\Delta_x J_x u_A+\Delta_y J_y u_B\right)^2\right]. \quad \text{(C9)}$$

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
