# Peer review of "Z2 order fractionalization, topological phase transition, and odd frequency pairing in an exactly solvable spin-charge ladder"

_SciPost Physics_

## Round 1 · Referee Report · Anonymous (Referee 1) · 2024-11-6

Strengths

1) Interesting phase with Z2 order fractionalization and odd-frequency pairing which is described non-perturbatively
2) Paper which is well written with step by step calculations

Weaknesses

1) The model is only studied in a very particular limit where the exactly solvable Hamiltonian is quadratic in terms of fermions
2) Odd-frequency, pair-density wave phases with composite order parameters have already been identified in many 1D lattice models (Kondo lattice, Kondo-Heisenberg chain models)

Report

This paper presents a two-leg ladder, a spin-charge ladder, where the 1D Kitaev p-wave superconductor model with dimerized hopping and pairing interacts with a 1D limit of the Kitaev’s honeycomb model via an anisotropic Kondo coupling. The model is directly inspired by the recent works of Coleman and Tsvelik on two and three-dimensional Kitaev-Kondo lattice models which harbor exotic physics such as odd-frequency pairing and order fractionalization with the formation of a bound state between electrons and Majorana fermions.
Here, the authors studied directly this phase in one dimension in the spin-charge ladder model which becomes exactly solvable in a specific limit through a mapping onto Majorana fermions.
The solution follows now a standard approach, by introducing four Majorana fermions for the spin chain, and two Majorana from the complex fermion of the superconducting wire following Kitaev's solutions of its honeycomb model and its famous 1D model. In a special limit of the coupling constants, the model becomes an exactly solvable model where Majorana fermions are coupled to static Z2 bosonic matter and gauge fields. The analysis reveals a phase with Z2 order fractionalization which stems from the condensation of the bosonic matter (represented by a pair of static Majorana fermions) and the Z2 gauge field (visons) is fully gapped. This “Higgs phase” is characterized by the formation of a bound state with fractional quantum numbers and odd frequency pairing.

The paper is well written, with correct calculations. The approach used in the paper is now standard and easy to follow. While the physical results are interesting, they are somewhat expected, given the established literature in 1D Kondo lattice models with odd frequency pairing and pair-density phases arising from the formation of composite bound states.
(see for instance Chen et al, Phys. Rev. Res. 6, 023227 (2024), Berg et al, Phys. Rev. Lett. 105, 146403 (2010), Zachar, Phys. Rev. B 205104 (2001).) Nevertheless, I recommend the manuscript for publication but suggest the authors address the following points:

Requested changes

1- The title is rather long and should be changed. In this respect, the emphasis of the topological transitions may be overstated, as such transitions in 1D are now standard and often well-understood through CFTs. In this paper, the transition has been studied by using spectral chiral index to count the Majorana zero modes. No details seem to be given on its universality class. Could the authors offer further comments on this ?

2- Fig. 4 is nice but difficult to follow since the authors do not consider the explicit difference between even and odd sites for the gauge fields uj (same problem with Fig. 2). The Majorana fermions involved in u2j and u2j+1 are different (see Eqs. 15,16) but on the figure it seems to result from the coupling between γtjγzj+1, which appears inconsistent with the definition (Eqs. 15,16) of the Z2 gauge field. Greater clarity on this point could be useful for the readers.

3- What happens for the Z2 ordered fractionalized phase away from half filling ?

Recommendation

Publish (meets expectations and criteria for this Journal)

---

## Round 1 · Referee Report · Anonymous (Referee 2) · 2024-11-11

Report

The authors derive an exactly solvable model for a mixed two-leg ladder with spin-1/2 degrees of freedom occupying one leg and spin-less fermions occupying the second leg and with Ising interaction along the rung. The authors consider the “Kitaev” set-up: along the top legs spins interact only in XX (YY) component along odd (even) legs; along the bottom leg the even-odd alternation is encoded through hoping and pairing amplitudes. The authors report Z2 order fractionalization with dual symmetry breaking. They claimed “intertwined” order parameters.

I find the manuscript rather pedagogical, but I am terribly missing a broader perspective. Neither in the discussion/motivation the authors explain the very specific model (I understand that it has to be exactly solvable, but the model in Eq.1 has 6 independent parameters!), nor they highlight an impact of their results. Phrases like “may generate novel quantum criticality” or “DMRG provides a powerful method to the future systematic study of the whole phase diagram” without any further details are too vague. I would like the authors to clarify the importance of their exact results in the broad context before I can asses whether this manuscript is more suitable for SciPost Physcis or SciPost Physics Core.

Major concern: The authors claim that they investigate the topological phase transition: they have introduced two order parameters and they claim a dual symmetry breaking, yet they write in the discussion section that “1+1-D Ising universality class is presumed”, as if it would be just one Z2 symmetry spontaneously broken at the transition. Do I miss something? From the information provided in the paper, my naive guess is that this transition with two Z2 symmetries broken on each side of it belongs to the 8-vertex universality class (see the chapter by Marcel den Nijs in Phase Transitions and Critical Phenomena vol. 12, 219, 1988; or recent Chepiga, Mila, Phys. Rev. B 107, L081106 (2023); and Roberts, Jiang, Motrunich, Phys. Rev. B 99, 165143 (2019)). I think, this has to be clarified.

In addition, I have a coupe of minor remarks that the authors might want to address to clarify the presentation:

- Sec.III: It would be great to provide the references to the local Majorana fermion representation (before Eq.5) or to say it explicitly that you are going to properly introduce this representation below.
-I am lacking an intuitive physical picture for the operators O_{+/-}. If the authors can think of some it would be very instructive to add those after Eq.81 or 82.
- I find all sketches for Majorana representations to be counter-intuitive. For instance, Fig.2: I would expect that orange bonds would connect lower pairs of black dots, this seems standard in the field and somehow refers to AKLT and Kitaev’ sketches.
- just avter Eq.24: “covention”.

Recommendation

Ask for major revision

---

## Round 1 · Referee Report · Anonymous (Referee 3) · 2024-11-14

Strengths

1. The authors suggest a new 1D model which shows "order fractionalization"

Weaknesses

1. The model is exactly solvable only at a specific point in the parameter space.
2. There is no coherent discussion of a physical picture of the results, or physical implications in e.g. possible experiments.

Report

I agree with the assessment of the second Referee.

The authors provide a lengthy pedagogical introduction in the first part of the paper, and an exact solution for a specific limit in the second part.

However, the discussion of the calculations/results, or their "broader perspective" an "physical impact" is missing. The lengthy and repetitive discussions are too vague (and use a lot of jargon) and it's often hard to understand the point the authors are trying to make, for example such as this sentence "However, the exactly solvable Hamiltonian
in Eq. (49) indicates the bosonic matter field and gauge
field are coupled by integrating the fermionic matter field."

I do not recommend this paper for publication in SciPost in the current form.

Requested changes

Suggestions:

1. The authors should clarify the physical picture, and provide a discussion of potential experiments

2. The model is only discussed in a specific exactly solvable limit, and it would be good to see the results of exact diagonalization beyond this limit, and the comparison of analytical results with the ED

3. Change J_x,J_y to J_1, J_2, and same for t_x, etc.

4. In the sentence "In the slave-particle theory, the electron is
fractionalized into spinon and holon carrying the charge
and spin degrees of freedom respectively", exchange the order of the words charge and spin.

5. Improve English, and correct mistakes

Recommendation

Ask for major revision

---

## Editorial Decision

awaiting_resubmission